# The Role of DNA Methylation in Stroke Recovery

**DOI:** 10.3390/ijms231810373

**Published:** 2022-09-08

**Authors:** Dong-Hee Choi, In-Ae Choi, Jongmin Lee

**Affiliations:** 1Center for Neuroscience Research, Institute of Biomedical Science and Technology, Konkuk University, Seoul 05029, Korea; 2Department of Medical Science, Konkuk University School of Medicine, Seoul 05029, Korea; 3Department of Occupational Therapy, Division of Health, Baekseok University, Cheonan 31065, Korea; 4Department of Rehabilitation Medicine, Konkuk University School of Medicine, Seoul 05029, Korea

**Keywords:** stroke, treatment, recovery, epigenetics, DNA methylation, neuroprotection, neurorepair

## Abstract

Epigenetic alterations affect the onset of ischemic stroke, brain injury after stroke, and mechanisms of poststroke recovery. In particular, DNA methylation can be dynamically altered by maintaining normal brain function or inducing abnormal brain damage. DNA methylation is regulated by DNA methyltransferase (DNMT), which promotes methylation, DNA demethylase, which removes methyl groups, and methyl-cytosine–phosphate–guanine-binding domain (MBD) protein, which binds methylated DNA and inhibits gene expression. Investigating the effects of modulating DNMT, TET, and MBD protein expression on neuronal cell death and neurorepair in ischemic stroke and elucidating the underlying mechanisms can facilitate the formulation of therapeutic strategies for neuroprotection and promotion of neuronal recovery after stroke. In this review, we summarize the role of DNA methylation in neuroprotection and neuronal recovery after stroke according to the current knowledge regarding the effects of DNA methylation on excitotoxicity, oxidative stress, apoptosis, neuroinflammation, and recovery after ischemic stroke. This review of the literature regarding the role of DNA methylation in neuroprotection and functional recovery after stroke may contribute to the development and application of novel therapeutic strategies for stroke.

## 1. Introduction

Globally, stroke is the primary cause of long-term impairment, which results in tremendous medical and financial problems. Ischemic stroke constitutes about 87% of all incident strokes [1,2]. Tissue plasminogen activator (tPA) has been approved as a treatment for cerebral infarction over the past two decades and continues to be actively utilized worldwide. Clinically, the treatment period for tPA is narrow, and it has side-effects, such as intracranial hemorrhage and neurotoxicity [1,3,4]. The pathological processes of ischemic stroke include excitotoxicity, oxidative stress, apoptosis, inflammation, mitochondrial dysfunction, and autophagy [1,3,4]. Various neuroprotective strategies are being developed to intervene in these pathological processes. Another treatment approach is the augmentation of neurorepair after stroke. Neurorestoration following stroke requires a series of highly interdependent processes, including neurogenesis, angiogenesis, axonal regrowth, and synaptic plasticity [1,5]. These processes work together and contribute to restore neural function. Neurorestorative therapies may, thus, have a significantly longer treatment time window than that of neuroprotective therapies [1,6].

In the central nervous system (CNS), epigenetic mechanisms are sensitive to both local and global in vivo environments; they act on vascular, systemic, and CNS-specific factors, and serve as key regulators of brain development, homeostasis, and neuroplasticity [7]. Control and modification of gene expression are necessary for cell survival and homeostasis, to sustain the biological state [8]. Epigenetic alterations may be implicated in the pathogenesis and recovery of various diseases via molecular and cellular mechanisms and are known to regulate cell survival and tissue repair in stroke [7]. Pathophysiological ischemic conditions can induce and/or repress specific genes, to induce protective and/or damaging pathways in cells [8]. In recent years, numerous studies have presented evidence supporting the importance of epigenetic modifications in ischemia-induced brain damage. Gene expression can be modified without DNA sequence alteration through epigenetic modifications [9], such as DNA methylation, histone modifications, and regulation by noncoding RNAs [9].

Accumulating research suggests that DNA methylation plays a multifaceted role in the various pathological mechanisms induced by cerebral ischemia [7,10]. Molecular studies performed using animal models of stroke have shown an increase in the amount of global DNA methylation after ischemic injury, and this phenomenon has been correlated with increased brain damage on the basis of experimental evidence [8,11,12]. Although studies on the effects of DNA methylation on the pathogenesis of stroke and the recovery mechanisms after stroke are ongoing, further research is needed.

Depending on whether aberrant DNA methylation can induce neuronal damage in stroke conditions, novel therapies can be developed on the basis of these mechanisms. Therefore, the goal of this review was to identify a new neuroprotective and accessible DNA methylation regulation technique that promotes neuronal recovery by investigating the mechanisms of neuronal death and neural repair via DNA methylation regulation after stroke reported to date. Our specific purpose was to investigate the correlation between DNA methylation and neuroprotection or neurorepair in treating post-stroke cerebral ischemia. Recent experimental studies have demonstrated changes in DNA methylation after stroke or ischemic conditions in rodent models and in vitro cellular experiments, indicating that they affect regulation of injury and promotion of stroke recovery.

Here, we review the present state of research on the role of DNA methylation in stroke treatment, including neuroprotection and neurorepair after stroke. 

## 2. Methods

This review focused on studies investigating the association between DNA methylation in stroke treatment. A literature search was carried out using the databases of Medline and PubMed without date and language restriction until May 2022. All articles were solicited from the databases using the searching terms “DNA methylation treatment”, “DNA methylation neuroprotection”, “DNA methylation recovery”, and “stroke”. A keyword search for “DNA methylation treatment/neuroprotection/recovery” was performed using medical subject headings (MeSH) terms and all field terms. The stroke keyword was searched using MeSH terms. The full search details for all databases can be found in the Appendix A. For screening and selection in the next stage, only full-text articles were considered for inclusion in further analyses. From a total of 67 articles identified in the searching process, 26 articles met the criteria for inclusion. 

Figure 1 is a flowchart of the literature searching, screening, and study selection, eligibility criteria, and inclusion of eligible articles from the literature. Up to 18 April 2022, we searched potential articles in the literature from the databases Medline and PubMed. The records identified from databases resulted in 67 studies. Seventeen duplicate articles were removed before screening and 50 studies were selected as the screened reports. A retracted article was excluded from the screened records. In second stage, clinical research papers and articles including mismatched outcomes and diseases were excluded due to eligibility. The literature published from 1 May 2000 to 18 April 2022 was searched to identify eligible articles for review. In the next step, two co-reviewers (D.H. and I.A.) independently screened all potentially relevant titles and abstracts for eligibility and, if necessary, checked full-text articles. Data extraction and assessment of neuroprotection or recovery after regulation of DNA methylation in stroke were independently evaluated by two authors (D.H. and I.A.). The data on neuroprotection and neural recovery evaluated the specific mechanisms involved in the process. In case of discrepancies between the two reviewers (D.H. and I.A.), a third reviewer (J.L.) was included in the selection process, and consensus was reached through mutual communication between the reviewers. As a result of the study according to the methodology, the final total number of articles included in this review was 26 studies.

## 3. DNA Methylation in Stroke

The DNA methylation-based epigenetic mechanisms responsible for increased cytosine–phosphate–guanine (CpG) island methylation transcriptionally inhibit both specific and global gene expression [8]. Various studies on the role of DNA methylation in inducing ischemic stroke have been published recently, and promoter DNA methylation has investigated in multiple genes [9,13]. Studies using animal models of ischemic stroke have shown that the total DNA methylation levels are generally elevated after ischemia, in association with increased DNA methyltransferase (DNMT) activity [9]. Increased DNA methylation after ischemic injury is reported to heighten the transcriptional repression of numerous genes, thereby increasing brain damage [8,11,12].

DNA methylation is carried out by the DNMT-catalyzed covalent conversion of cytosine to 5-methylcytosine (5mC), a cytosine derivative with an additional methyl group at the C5 position [14]. This procedure modifies DNA stability and accessibility, thereby regulating gene expression [15,16]. Hydromethylation (5hmC) is another epigenetic mechanism that adds a hydroxymethyl group to 5-methylcytosine bases [15,16].

In a study aiming to verify the role of DNMT in ischemic injury, inhibition of DNMT expression and activity through pharmacological or genetic manipulation was found to suppress ischemic injury and improve functional brain recovery [7,12,17]. Thus, DNA methylation is a critical feature that may affect the outcomes of brain injury. DNA methylation is an epigenetic mechanism mediated by DNMTs and involves the addition of methyl groups to DNA bases. This process results in chromatin condensation and altered gene expression [9]. Two groups of DNMTs, i.e., maintenance and de novo enzymes, have been identified [18]. The earliest and most prevalent DNMT, DNMT1, is referred as a “maintenance methyltransferase” because it can methylate the base pair of hemi-methylated DNA. DNMT3a and DNMT3b are de novo methyltransferases that methylate originally nonmethylated cytosines [19].

Furthermore, the methylcytosine dioxygenase, ten-eleven translocation [20], oxidizes the methyl group of 5mC to produce 5hmC [16,21]. The three major isoforms (TET1–3) of TET are widespread in the brain and activate genes whose expression is suppressed by DNA methylation [22]. TET can further oxidize 5hmC to generate 5-formylcytosine (5fC) and 5-carboxylcytosine (5caC) modifications sequentially, which can be directly cleaved to yield unmethylated cytosine [22,23].

In addition to DNMT and TET, which regulate the methylated sequence of the promoter regions in genomic DNA, other factors including the methyl-CpG-binding domain (MBD) proteins bind to the methylated sequence of genomic DNA to regulate transcription regulatory factors and control gene expression [24]. Among the methyl-DNA-binding family genes, five independent genes, i.e., those encoding the methyl-CpG-binding protein 2 (*MeCP2*) and *MBD1–4*, have been cloned to date [24,25]. Repression of transcription can be achieved by methyl-DNA-binding factors, either through direct binding to methylated promoters or through the recruitment of larger heterologous protein complexes to methylated genomic sites [24,25]

Several studies on humans and animal models related to stroke have identified aberrant DNA methylation patterns [13]. Animal and cell experiments have revealed hypermethylation at specific gene-promoter sites, as well as the effect of treatment with DNMT inhibitors [9,12,26,27,28]. Moreover, hyper- or hypomethylation at different gene promoters has been reported in patients with stroke [29,30]. The outcomes of these methylation conditions indicate a role for methylation in inducing cerebral infarction pathologies or regulating the recovery mechanisms after stroke [9].

In the next section, we discuss the neuroprotective and neurorepair mechanisms activated after modulating the alterations in DNA methylation after ischemic stroke.

## 4. DNA Methylation in Neuroprotection after Stroke

Here, we review the role of DNA methylation in neuroprotection through the regulation of excitotoxicity, oxidative stress, mitochondrial dysfunction, blood–brain barrier (BBB) disruption, apoptosis, and inflammation, all of which are pathophysiological events [1,2,4,31] that occur after stroke.

### 4.1. Excitotoxicity

Glutamate is the driving factor of ischemia-induced excitotoxicity. Glutamate is the most prevalent free amino acid in the CNS and functions as an excitatory neurotransmitter [32,33]. As the excitatory Na^+^-dependent amino-acid transporter (EAAT2), which acts as the astrocyte glutamate transporter, takes up vast amounts of glutamate, dysregulated EAAT2 expression is implicated in the pathogenesis of neurological disorders with prominent excitotoxic components [34]. Hypermethylation of the classical CpG island of the *EAAT2/GLT1* gene promoter, as revealed in a human glioma cell line, occurs locally and prevents glutamate uptake, which can lead to energy deficiency due to reduced glutamine production, resulting in neuronal damage [34,35]. In contrast, the same promoter region is hypomethylated in normal human brain tissue expressing EAAT2 [34]. During hypoxic ischemia, the activity of DNMT1 and DNMT3A is increased and MBD2 is downregulated in astrocytes, resulting in increased hypermethylation of the CpG island of the *EAAT2/GLT1* gene promoter in astrocytes. In several human glioma cell lines, the *EAAT2* promoter is hypermethylated, which is correlated with low EAAT2 protein expression [34,35,36].

Multiple studies have indicated that EAAT2 upregulation confers neuroprotection during ischemia [37]. GFAP-driven EAAT2 expression in astrocytes was reported to enhance neuroprotection in murine hippocampal slice cultures after moderate oxygen–glucose deprivation [37,38]. Several animal models of ischemia are known to be protected by ceftriaxone-triggered *EAAT2* expression, which increases EAAT2 protein expression and activity [37]. The studies reported to date have been insufficient to investigate the direct effect of DNA methylation changes on EAAT2 upregulation, which demonstrates neuroprotective effects under cerebral infarction conditions. However, this effect may be explored in future by applying hyper-/hypomethylation of EAAT2 as shown in human glioma cell lines.

Within the mammalian CNS, γ-aminobutyric acid (GABA) is the primary neurotransmitter with an inhibitory effect. GABA receptor-mediated responses can result in either membrane depolarization or hyperpolarization, depending on the intracellular Cl^−^ concentration [27]. In turn, Cl^−^ concentration is regulated by two Cl^−^ cotransporters, Na–K–Cl cotransporter (NKCC) 1 and K^+^–Cl^−^ cotransporter type 2 (KCC2), and is linked to GABA activity. Therefore, NKCC1 increases the depolarization response early in development, and KCC2 increases hyperpolarization responses during the postnatal growth phase. Some pathological conditions that cause nerve damage, including axonectomy, ischemia, and excitotoxicity, can alter the intracellular Cl^−^ concentration, even in mature neurons [27]. Such alterations result from either KCC2 downregulation [27] or NKCC1 upregulation. In an experiment using rat cortical section cultures, 5-aza-2′-deoxycytidine (5-Aza-dC), an inhibitor of DNA methylation, was found to increase NKCC1 mRNA and protein expression, suggesting that the DNA methylation mechanism is involved in NKCC1 expression [27]. In contrast, upregulation of KNCC1 mRNA and protein was observed after ischemia or cerebral infarction in oxygen–glucose-deprived cultured cells and under ischemic conditions in animals. DNA methylation in the *KNCC1* gene promoter region is decreased in hypoxic–ischemic conditions [27]. Furthermore, methylation of the NKCC1 promoter was decreased in the damaged cortical region of mature mice induced with middle cerebral artery occlusion (MCAO), and. as a result, the expression of KNCC1 mRNA and protein was increased [27].

### 4.2. Oxidative Stress

Oxidative stress is among the most potent causes of epigenetic dysregulation in ischemic stroke. During ischemic stroke, glutamate-induced excitotoxicity results in oxidative stress, which is inextricably linked with energy deficit [35]. The oxidative stress occurring after experimental stroke in animal and cellular models has been found to cause epigenetic changes in neurons, glial cells, and vascular endothelial cells, as well as interactions between cells in redox states [39]. In turn, the altered redox status affects DNA methylation, and oxidation of 5mC to 5hmC inhibits the interaction between proteins that bind to methylated-CpG, resulting in an altered epigenetic state [35]. Epigenetic processes are, thus, controlled by DNA methylation and oxidative damage.

A byproduct of oxygen radicals in DNA, 8-hydroxyl-2′-deoxyguanosine (8-OH-dG) is produced by the attack of hydroxy radicals generated under oxidative stress, and is quadrupled in the urine of patients with stroke and in ischemic animals [35,40]. In hemi-methylated DNA, increased 8-OH-dG interferes with the methylation of the adjacent cytosine on the nascent DNA strand, which may result in incorrect coding. Steric interference by 8-OH-dG-modified guanine inhibits the high-affinity binding of DNMTs required for nucleotide complex formation, leading to hypomethylation [35,40,41]. Further, 8-OH-dG is responsible for guiding the removal of base sites in the base excision repair machinery. Therefore, when 8-OH-dG levels increase because of oxidative damage, aberrant DNA modification of gene promoter regions in human and mouse cells induces an epigenetic process [35].

The oxidative stress induced by stroke is characterized by an imbalance between increased reactive oxygen species (ROS)/reactive nitrogen species (RNS) production and antioxidant activity, which results in oxidative modification of proteins, lipids, and DNA [39]. Oxidative damage to DNA normally inhibits DNMT1-mediated methylation, resulting in methylation loss in progeny cells. These alterations are positively correlated with oxidative stress [8,39,42]. Production of ROS/RNS (e.g., H_2_O_2_, nitric oxide) oxidatively converts 5mC to 5hmC. The DNA methylation pattern is altered by inhibiting the binding of DNMT1 and methyl-binding protein to DNA [8,39,41,42]. Various modifications of 5mC occur because of the DNA damage caused by ROS after a stroke, resulting in increased 5hmC levels and inappropriate modification of 5mC [17,39,43]. In the MCAO mouse model, 5mC and 5hmC levels were found to be elevated at different time periods after ischemic reperfusion injury, and 5hmC levels were found to be considerably elevated in human blood following ischemic stroke [17].

Reactive inflammatory molecules induce altered cytosine, including 5-chlorocytosine that carries halogens at the 5-position in cytosine and oxidation-induced products such as hydroxycytosine, 5hmC, and 5-(1-propynyl)-cytosine, which cause lethal damage to cells [43]. The human maintenance protein DNMT misrecognizes 5-chlorocytosine, and inappropriately methylates previously unmethylated CpG sites [42,43], leading to the binding of methylated DNA-binding proteins [43]. Myeloperoxidase (MPO)-generated superoxide presented on infiltrating neutrophils, activated microglia, neurons, and astrocytes in the ischemic brain mimics 5mC with 5-chlorocytosine. MPO activation catalyzes the interaction between chloride and H_2_O_2_, to form HOCl. MPO controls oxidative stress by increasing the production of ROS and RNS, and by adjusting the polarization of microglia and neutrophils, as well as signaling pathways related to inflammation [44]. Nucleotides in 5-chlorocytosine-modified DNA induce inappropriate DNMT1 methylation within the CpG sequence, leading to gene silencing [17,43,44].

Recent research has identified vitamin C as an epigenetic regulator that acts by activating the TET enzymes involved in the production of 5hmC in DNA [45].

Ascorbate (vitamin C) is an essential brain antioxidant chemical [46]. The sodium-dependent vitamin C transporter 2 (SVCT2) transports ascorbate into neurons and the brain, resulting in the intracellular accumulation of ascorbate against a concentration gradient [46]. The oxidized form of ascorbate, dehydroascorbic acid, is carried by members of the glucose transporter (GLUT) family. Once inside the cell, it is quickly converted to ascorbate [46]. Poststroke therapy using ascorbate (reduced form), but not dehydroascorbate (oxidized form), has been found to boost TET3 activity and 5hmC levels, as well as decrease the infarct size in response to focal ischemia [45,47]. In addition, ascorbate promotes motor and cognitive function recovery by increasing TET3 and 5hmC, and decreases oxidative stress and apoptosis in cortical peri-infarct neurons [22,45]. Collectively, the results of these studies demonstrate that the mechanism via which TET3 increases 5hmC under oxidative damage after stroke is also involved in protecting the brain from ischemic damage. This evidence establishes a mechanistic connection between oxidative damage and epigenetic changes through the chemical modification of DNA bases and modified DNA–protein interactions [39]. Ischemia or oxidative stress are known to affect global DNA methylation in neuronal cells, and DNMT inhibitors may alleviate the brain damage caused by ischemia or oxidative stress [39].

### 4.3. Mitochondrial Dysfunction

Although controversial, recent research indicates that dysfunction in mitochondrial DNA methylation may cause several diseases [48]. In particular, understanding the regulatory mechanisms of mitochondrial epigenetics involved in the pathophysiological process of ischemic stroke may be important.

Mitochondria are a significant source of ROS in ischemic stroke. Ischemic damage causes swelling of mitochondrial matrix, blockage of ATP production, and overproduction of reactive oxygen species, all of which contribute to mitochondrial membrane leakage [1,49]. In turn, opening of mitochondrial permeability transition (MPT) pores causes ischemic damage [1,49]. ROS serve as signaling molecules that transmit extracellular input to subsequent signaling cascades via epigenetic changes, which allow cells to adapt to the varying availability of bioenergetic sources [35].

The mitochondrial dysfunction occurring in ischemic stroke is caused by increased MPT, decreased oxygen uptake, and a subsequent decrease in ATP synthesis induced by oxidative stress. Increased mitochondrial oxidative stress is reported to be positively associated with increased matrix metallopeptidase (MMP)-9 activity, as well as autophagy, fusion, and division in ischemic mitochondria, leading to mitophagy and extracellular matrix (ECM) damage [48]. After cerebral ischemia, MMP-9 activity is enhanced, resulting in accelerated matrix degradation, BBB disruption, and infarct enlargement during stroke. Tissue inhibitors of metalloproteinase 1 and 2 (TIMP-1 and TIMP-2) inhibit MMP-9 [48]. Thus, the equilibrium between MMPs and TIMPs is crucial for correct ECM remodeling, as well as various developmental and morphogenetic processes [48].

A recent animal study on stroke revealed increased mitochondrial MMP-9 activity and decreased TIMP-2 expression after ischemic cerebral infarction [48]. DNMT activity evaluation in the mitochondria using 5mC and TIMP-2 methylation assays revealed that, after cerebral infarction, the mRNA and protein expression of DNMT1 and DNMT3a was increased, DNMT activity was increased, and methylation of TIMP-2 DNA was increased; these effects were reversed by 5-aza-dC treatment [48]. As an important mechanism for inducing mitochondrial dysfunction after cerebral infarction, increased mitochondrial MMP-9 activity is suggested to be caused by increased DNMT expression and activity, as well as decreased TIMP-2 expression, which is triggered by the increased methylation level of the *TIMP-2* promoter region. Therefore, regulation of TIMP-2 expression by DNMT inhibitor treatment in ischemic stroke may contribute to neuroprotective mechanisms by preventing the mitochondrial dysfunction that occurs after cerebral infarction. In a study on the demethylation mechanism after ischemic injury, an increase in mitochondrial TET2 expression resulted in an increase in mitochondrial DNA 5hmC without a change in the level of 5mc, and the increased 5hmC increased the mRNA level of mitochondrial genes. The gene includes the subunits (ND2, ND3, ND4L, ND5, and DN6) of complex 1 (NADH: ubiquinone oxidoreductase) and the COX3 subunit of complex 4 (cytochrome c oxygenase) [50,51].

### 4.4. Blood–Brain Barrier (BBB) Disruption

The BBB is a dynamic barrier comprising an interdependent network of brain capillary endothelial cells capable of barrier functions. In the BBB, endothelial cells form tight junctions (TJs) that limit the amount of material that can pass through [52]. Local neuronal function is coupled with local cerebral blood flow by the neurovascular unit (NVU), which consists of endothelial cells, a basement membrane, pericytes, astrocytes, and microglial cells. Moreover, the NVU is responsible for regulating the transport of bloodborne molecules across the BBB [52].

Altered DNA methylation is related to both BBB injury and restoration in the NVU, indicating a biphasic pattern of post-stroke DNA methylation [53]. In a mouse model, MCAO induced hypermethylation of the *Timp-2* promoter [53]. Reduced TIMP-2 expression is related to increased MMP-9 activity and BBB permeability, whereas DNMT inhibition decreases BBB permeability [48,53]. MCAO-induced *Timp-2* promoter hypermethylation results in decreased TIMP-2 activity and increased MMP activity, both of which contribute to TJ proteolysis. Consistent with these findings, both pharmacological and genetic suppression of DNMT were proven beneficial in reducing the symptoms associated with stroke [12,48,53].

Furthermore, in vitro and in vivo model experiments confirmed that disruption of the BBB can induce changes in DNA methylation and affect the expression of DNMT3b, a DNA methyltransferase enzyme [54]. Increased homocysteine (Hcy) after ischemic stroke is a major factor that induces destruction of the BBB [54]. DNMT3b protein was downregulated, and MMP-9 protein expression and activity were increased in an experimental model of BBB destruction caused by an increase in Hcy [54]. The role of DNA methylation in MMP-9 expression and activity was confirmed by the decrease in increased MMP-9 levels with an increase in Hcy induced by treatment with the DNMT inhibitor, 5-aza-dC [54].

### 4.5. Apoptosis

Apoptosis may play a significant role in neuronal death after acute brain ischemia [55]. Several hours or days after the beginning of severe brain ischemia, apoptosis may occur in the ischemic penumbra, whereas necrosis occurs in the core area of the ischemic brain within the first few hours after injury [55,56]. The simultaneous presence of apoptosis and necrosis markers in the same cell after ischemia suggests that multiple death programs are activated simultaneously [55,56].

In a study by Endres et al. [12], the role of DNMT in apoptosis induced by ischemic stroke was examined; they found a significant increase in the DNA methylation of wildtype (*Dnmt+/+*) mice in a MCAO model, whereas transgenic *DnmtS/+* mice showed no changes. Analysis of the infarct volume and viable cell number after MCAO indicated a decrease in infarct volume and an increase in viable cell number in the *DnmtS/+* mice. Furthermore, a decrease in lesion size was observed after treatment with 5-aza-dC, a DNA methylation inhibitor, in MCAO-induced wildtype (*Dnmt+/+*) mice [12,39]. The role of DNA methylation in ischemic brain injury was also demonstrated in a mouse ischemic brain injury experiment using zebularine, another DNA methylation inhibitor; zebularine treatment yielded a decrease in cerebral infarct volume and apoptosis [26].

In the MCAO animal model, DNA methylation is elevated in neurons located around the cerebral infarction at 24 h after MCAO and reperfusion, which is attributed to the increased activation of DNMT3a. An increase in DNMT3a was observed after *N*-methyl-d-aspartate (NMDA) treatment, and this increase was reversed by treatment with a DNMT inhibitor. These findings indicate that DNA methylation is an early event in neuronal damage, and that inhibiting DNA methylation may be an effective treatment for stroke [57]. In another animal model of photothrombotic stroke (PTS), the level of DNMT1 was increased in the nuclear fraction of neurons and in the cytoplasmic fraction of astrocytes in the penumbra tissue at 24 h after PTS induction; furthermore, apoptosis was induced. Inhibition of DNMT1 by 5-aza-dC protected against PTS-induced apoptosis in the penumbra at 4 days, as well as 24 h after PTS [58]. The effect of altered DNA methylation using 5-aza-dC on neuronal apoptosis was also confirmed in an experiment with mouse hippocampus-derived neuronal HT22 cells. A flow cytometry analysis showed that 5-aza-dC treatment reduced early apoptosis, while also increasing apoptosis in the later stages. At this time, decreased mRNA and protein expression of DNMT1 and DNMT3a was observed after 5-aza-dC treatment, indicating that HT22 apoptosis is associated with altered expression of DNA methylation regulators DNMT1 and DNMT3a [59]. Thus, downregulation of increased DNMT in ischemic cerebral infarction is an important neuroprotective mechanism that reduces infarct volume and apoptosis after ischemic injury [12,26,39,57,59].

Prenatal hypoxia downregulated the mRNA and protein expression of glucocorticoid receptor (GR) in the hippocampus of newborn rats and eliminated the neuroprotective effects of dexamethasone in the brains of canines. This reduction in GR expression results from elevated DNA methylation, reduced binding of Egr-1 and Sp1 to the promoters of exons 17 and 111 of *GR*, and lower transcript levels of *GR* exons 17 and 111. These findings indicate that gestational hypoxia induces epigenetic inhibition of *GR* gene expression in the developing brain, culminating in increased susceptibility of the brain to hypoxic–ischemic damage in newborn mice [60]. Mouse brain gene expression and DNA methylation profiling after ischemic preconditioning revealed dramatic changes in genomic DNA methylation, which in turn altered the gene expression profile; *Arid5a*, *Nptx2*, and *Stc2* were upregulated, indicating a neuroprotective function [61]

The involvement of DNA methylation in gene expression changes that emerge after ischemic stroke has been studied using fetal hypoxia, neonatal hypoxia–ischemic brain injury, and in vitro oxygen–glucose deprivation induction experiments [60,61,62]. Suppression of DNA methylation after neonatal hypoxic–ischemic brain damage is reported to confer resistance to ischemia, in addition to specific inhibition of histone deacetylation [12,62]. This may be associated with the upregulation of protective or antiapoptotic genes, such as *c-fos*, *bcl-2*, and superoxide dismutase (*SOD)*, or the downregulation of deleterious or proapoptotic genes, such as *bad* and *bax* [12,62].

Estrogen receptor alpha (ERα), known to play an important role in estrogen-induced neuroprotection, was significantly increased in terms of mRNA and protein in MCAO-induced female mice and mice. An epigenetic mechanism is involved in this process. After ischemia, decreased methylation was induced at various loci of the ERα promoter, and MeCP2, a methylated binding protein, was dissociated to increase the expression of ERα mRNA [7,63,64].

New evidence indicates that TET and 5hmC exert neuroprotective roles. In studies examining the role of type-specific TET in neuronal damage, TET1 activity protected neurons against ROS and caspase-3-dependent neurotoxicity, whereas neurons from *Tet-1*-knockout animals were more sensitive to oxidative injury [47]. Furthermore, TET2 and 5hmC depletion reduced neuronal survival, damaged hippocampal neuron growth, and decreased cognitive function in adult mice [47,65]. In contrast, reconstitution of the lost TET2 suppressed cognitive decline in the aged brain [47,65]. TET3 also plays an important role in the DNA damage repair pathway, triggers axonal regeneration after injury, and promotes synaptic plasticity in learning and memory processes [47]. Furthermore, TET3 provides a key protective mechanism against neuronal cell death in the brain by regulating the expression of genes associated with neurodegenerative diseases in the lysosomal and autophagic pathways [47].

Ascorbate acts as an epigenetic regulator because of its capacity to control TETs, the DNA enzymes responsible for converting 5mC to 5hmC. Ascorbate, which functions as a cofactor for TETs, upregulates both the TET activity and the levels of 5hmC [45]. Application of focal ischemia in adult mice increases cortical 5hmC from 5 min to 3 days after reperfusion; however, inhibition of TET3, an enzyme that produces 5hmC, increases brain swelling, infarct size, and motor dysfunction [45]. Previous reports have identified neuroprotective genes linked to 5hmC and modulated by TET3 [22,47].

In other studies on animal models with four-vessel occlusion, as the timing of hippocampal neuron death caused by ischemic insult in the hippocampus varies according to the subfield (CA1, CA2, CA3, and dentate gyrus, DG), the spatial and temporal expression of the MBD family was investigated and found to be correlated with neuronal death [24,39]. In particular, the CA1 subfield within the hippocampus is known to contain a neuronal population that is highly susceptible to ischemic conditions [24]. After four-vessel occlusion, MBD3 expression was found to be significantly decreased in the CA1 subfield at 3 h after ischemia, particularly at 24 h after ischemia. Furthermore, MBD2 expression was elevated for up to 6 h following ischemia, whereas MBD1 and MeCP2 levels were elevated for up to 24 h [24,39]. MBD2 and MBD3 coexist within the MeCP1 heterologous transcriptional repression complex, [24,66]. Intracellularly, they form homodimers or heterodimers, depending on the relative abundance of each molecule [24,67]. Therefore, when the ratio of MBD2 to MBD3 is significantly altered, the heterolog-to-homolog balance of MBD2 may be altered. Confirming these changes, the MBD2/MBD3 ratio and neuronal cell death were significantly increased in the hippocampal CA1 area at 24 h after cerebral infarction, whereas there was no change in apoptosis or the MBD2/MBD3 ratio in the DG [24]. These findings may explain the neuronal vulnerability in CA1.

To understand the changes occurring in the gene transcription complex according to the changes in MBD expression, with the exception of direct regulation of DNA methylation after cerebral infarction, the increase in the MBD2/MBD3 ratio and the expression of MeCP2 and MBD1 were confirmed in the CA1 region of the hippocampus after cerebral infarction [7,24]. These results indicate that modulation of MBD family factors after ischemic brain injury could be applied as a neuroprotective strategy.

### 4.6. Inflammation

Neuroinflammation contributes significantly to cell death, neuronal cell damage, and brain dysfunction in patients with stroke [68]. The epigenetic mechanisms involved in post-stroke pathophysiology include differential regulation of gene expression, particularly of genes involved in brain inflammation and post-stroke remodeling [68]. In this section, we discuss recent findings on the role of DNA methylation in regulating poststroke neuroinflammation and potential epigenetic targets that can be evaluated in the development of stroke therapies.

Analysis of DNA methylation after ischemic stroke has mainly revealed hypermethylation in patients with stroke and in animal experiments; however, hypomethylation has also been detected [9,68]. In this context, low blood levels of DNA methylation were found to be associated with a higher risk of ischemic stroke. This differential DNA methylation is likely to be involved in neurotoxicity and neuroprotection, resulting in increased or decreased in DNA methylation depending on the temporal aspect of stroke progression or region of the genome [68].

The long-interspersed nucleotide element 1 (LINE-1), one of the repetitive translocation elements that constitute approximately 55% of the human genome, is heavily methylated under normal conditions [29]. Upon LINE-1 demethylation, its activity as a retro-transposable sequence increases, resulting in genome modification by gene insertion and/or homologous recombination [29]. Demethylation of the *LINE-1* gene can increase the transcription of genes that contain LINE-1 sequences in their regulatory regions [29,69]. Recent human studies have found that LINE-1 expression mediates ischemic heart injury [9,29,70]; moreover, global or *LINE-1* DNA methylation was found to be decreased in the atherosclerotic lesions of patients with atherosclerosis [29]. In contrast, hypomethylation of the *LINE-1* DNA sequence may exacerbate stroke injury in relation to elevated levels of the vascular cell adhesion molecule 1 (VCAM-1), a protein that promotes vascular–immune cell interactions and mediates atherosclerosis, [29,68].

The tumor necrosis factor receptor (TNFR)-associated factor (TRAF) family, which is involved in inflammatory signaling events in endothelial cells, has also been implicated in inflammatory vascular diseases, such as atherosclerosis [30,68]. In particular, the TRAF3 protein participates in the signaling of CD40 and TNFRs, which are important for immune response activation. As low methylation of the 37 CpG site of TRAF3 correlates with increased platelet aggregation in inflammatory vascular disease, these factors may increase the recurrence of ischemic stroke [30,68,71]. The increased protein level of the protein phosphatase magnesium dependent 1A (PPM1A), one of the inflammatory control gene, is caused by induction of the hypomethylation of the PPM1A promoter, and the increased PPM1A protein has been found to increase the stroke recurrence by involving the regulation of the transforming growth factor-β (TGF-β) pathway [72].

Furthermore, methylenetetrahydrofolate reductase (MTHFR) is involved in initiating inflammatory reactions and is, therefore, associated with an elevated risk of stroke [7,73]. Methionine is a methyl donor for DNA methylation, and MTHFR is an important enzyme in Hcy metabolism that catalyzes the regeneration of methionine [74]. Elevated homocysteine levels can induce an excessive inflammatory response in the cerebral region [68]. Furthermore, high *MTHFR* promoter methylation levels are suggested to lower the risk of ischemic stroke [7,75]. These findings support the possibility of using the methylation status of *MTHFR* promoter as a predictive biomarker for ischemic stroke; however, additional studies are required to confirm this hypothesis. Moreover, regulating *MTHFR* promoter methylation may enable the development of novel therapeutic strategies. In a post-stroke depression (PST) model, demethylation was reduced by downregulation of TET2 in the lymphocyte enhancer 1 *(Lef1)* gene promoter, resulting in decreased *Lef1* gene expression. Reduction in 5hmC by suppressed TET2 decreases the activation of the Wnt/β-catenin/LEF1 signaling pathway, thereby increasing the expression of inflammatory factors, leading to PSD [76].

## 5. DNA Methylation in Neurorepair after Stroke

Immediately after stroke, enhanced plasticity occurs, similar to that observed in neurodevelopment [72], which encompasses the formation of new cells and blood vessels, growth and development of new axons, and formation and regulation of new and established synapses [72]. Although the mechanism via which the fully developed brain can immediately enter new stages of growth and progress in response to injury remains unclear, a key component of epigenetic regulation likely mediates this system [72].

The epigenetic mechanisms involved in post-stroke recovery include neurogenesis, angiogenesis, gliogenesis, axon growth, and synaptic plasticity [77]. In addition to being a major contributor to developing and maintaining neural networks, synaptic plasticity is a key mechanism in the innate process of post-stroke recovery [77,78]. Among the epigenetic recovery mechanisms induced after stroke, DNA methylation may play a significant role in post-stroke ischemic injury and recovery [77].

Several studies have investigated the significance of DNMTs in synaptic plasticity mechanisms [77]. Suppression of DNMT activity has been found to limit the long-term potentiation of hippocampal regions, resulting in reduced promoter methylation of the *BDNF* gene, which is implicated in synapse formation [77]. *Dnmt*-deficient mice demonstrate memory impairment and decreased synaptic plasticity in the hippocampal region [77].

Therefore, we reviewed the effects of epigenetic DNA methylation on the mechanisms of neurogenesis, angiogenesis, axonal growth, and synaptic plasticity involved in recovery after stroke.

### 5.1. Neurogenesis

During recovery from a stroke, the peri-infarct region of the brain is essential for the neuronal regrowth and restoration of function [79]. Neurogenesis directly impacts axonal restoration, dendrite growth, and synaptic connections [79]. Reestablishment of some of the damaged neural circuitry also significantly improves neural function [79]. Moreover, stroke induces neurogenesis in adults and even in elderly patients [79]. In a MCAO mouse model, removal of newly generated neuroblasts from the ischemic brain decreased nerve function recovery and induced cognitive deficits [79], suggesting that poststroke neurogenesis also affects neurological function [79].

DNA methylation is regulated by DNMT, TET, and MBD proteins during neural development and function [79]. Knockdown studies of *Dnmt1*, *Dnmt3a*, *MBD1*, and *MeCP2* have revealed that direct epigenetic regulation of various neuronal genes significantly affects both proliferating neural progenitor cells (NPCs) and differentiated neurons [79,80]. 

High levels of Hcy, a significant risk factor for ischemic stroke, have been shown to inhibit neuronal regeneration. Related studies have demonstrated that Hcy inhibits the self-renewal capacity of neural stem cells (NSCs). This was confirmed by the Hcy-induced reduction in the number of new immature neurons co-expressing doublecortin (DCX) and 5-bromo-2′-deoxyuridine (BrdU) and of newly regenerated mature neurons co-expressing neuronal nuclear antigen (NeuN) and BrdU in the hippocampus [81]. Neurogenesis levels are frequently assessed using DCX, a microtubule-associated protein expressed in developing neurons and migrating neuroblasts [81,82]. NeuN, a recognized neuron-specific marker, is used to assess newly formed hippocampal neurons, and BrdU is used to identify newly divided cells [81,83]. When DNMT activity, global 5mC levels, and intracellular 5mC levels in newly generated cells were measured in Hcy-treated MCAO ischemic brains to confirm the role of DNMT and DNA methylation in Hcy-induced nerve regeneration, decreased DNMT activity, reduced total methylation levels, and reduced numbers of neurons co-expressing 5mC and NeuN or DCX and 5mC were observed. A reduction in DNMT activity, which is mainly mediated by *S*-adenosylmethionine (SAM) and *S*-adenosylhomocysteine (SAH) concentrations, may produce Hcy-induced DNA hypomethylation [81,84,85]. According to these studies, the neurogenesis mechanism in stroke recovery can be enhanced by regulating DNMT activity, which is inhibited by the increased Hcy levels after stroke.

Studies examining the effect of MBD1 on adult neurogenesis have revealed that neural stem cells from *Mbd1*-knockout mice exhibit decreased neuronal cell development and increased genetic mutations. Moreover, the dentate gyrus of the hippocampus shows a significant decrease in long-term reinforcement in adult *Mbd1*-knockout mice, along with decreased neurogenesis and impaired spatial learning. Overall, these results suggest that DNA methylation preserves the cellular genome stability and is crucial for healthy neural progenitor cell and brain activity [79,80].

A study validating the functional role of TET1 in adult neurogenesis showed that *Tet1*-knockout mice had significant defects in the maintenance of short-term memory and learning delay, indicative of cognitive impairment, together with a significant decrease in the number of NPC cells. Furthermore, the promoters of gene sets involved in neural progenitor cell proliferation were hypermethylated in *Tet1*-knockout mice [79,86,87,88]. With a decrease in the number of NPC cells, 10-fold higher CpG methylation was observed in *Tet1*-knockout mice compared with that in normal mice, and the resulting decrease in overall gene expression was remarkable. In particular, hypermethylation was observed in the promoters of *Galanin* and *Ng2*, which are involved in neural progenitor cell proliferation, and the expression of these genes was reduced concomitantly [79,86,87,88]. These results indicated that adult neurogenesis can be significantly regulated by TET1 through DNA demethylation [86,88]. Therefore, to promote neurogenesis after stroke, regulation of DNA methylation by modulating TET1 expression can be considered a potential therapeutic mechanism.

In addition to oxidative 5mC demethylation by TET, three other mechanisms of 5mC demethylation have been proposed: (i) replacement of the methylated or oxidized base with unmethylated cytosine via base excision repair (BER) [23,89], (ii) deamination of 5mC after BER [89], and (iii) removal of 5mC by nucleotide excision repair (NER) [89,90]. Growth arrest and DNA damage-inducible 45 (Gadd45) family proteins are known to regulate active DNA demethylation [89,90,91,92]. A new mechanism was identified involving Gadd45a, a nonenzymatic element that induces DNA demethylation [90,91]. This mechanism may be triggered by NER because of the interaction between Gadd45a and the NER endonuclease, XPG, as demethylation requires the presence of transcriptionally bound NER [91]. In the nervous system, Gadd45a is mainly activated by damage signals, whereas Gadd45b is involved in neural activity involving induction of immediate early genes [90,91]. Synchronized activation of DG neurons during adult neurogenesis firmly induces Gadd45b and results in the maintenance of increased neuronal growth in the hippocampus. In contrast, *Gadd45b*-deficient mice demonstrate impaired neuronal activity-induced cell growth; furthermore, *Gadd45b*-knockout neurons are unable to demethylate DNA, thus failing to express brain growth and trophic factors such as fibroblast growth factor 1 (Fgf1) and brain-derived neurotrophic factor (Bdnf) [72]. In adults, Gadd45b is thought to regulate neurogenesis in response to external signals by actively demethylating DNA [72,89,92]. According to these findings, DNA demethylation control via regulation of Gadd45 in the neurogenesis mechanism involved in neuronal recovery after stroke can be considered as a therapeutic application.

### 5.2. Angiogenesis

Inflammation-mediated angiogenesis is an important physiological process in various diseases. However, increased and decreased angiogenesis have different effects on these diseases [68]. Typically, in malignant tumors, antiangiogenic and antitumor mechanisms are activated by increased expression of an endogenous antiangiogenic factor, thrombospondin 1 (THBS1). Angiogenesis is induced in response to ischemia [28]. In diseases such as stroke, THBS1 induces and increases angiogenesis by silencing theTHBS1 expression induced by hypermethylation of the *THBS1* promoter region [68,71,93,94]. In contrast, increased THBS1 expression may terminate the angiogenesis involved in stroke recovery [28]. Experimentally, a decrease in THBS1 mRNA and protein levels has been associated with hypermethylation of the *THBS1* promoter in the cerebral endothelial cells of hypoxia-exposed mice. Ischemia-induced methylation of the *THBS1* promoter was reduced by reoxygenation, which also increased the THBS mRNA and protein levels [7,28,52,72,94]. These results imply that methylation or demethylation of the *THBS1* promoter could control the process of post-stroke angiogenesis [28,52,94]. Conversely, THBS1 can stimulate the growth of new blood vessels in several experimental models [95,96,97]. THBS1 exerts an angiogenic effect owing to its ability to promote the function of inflammatory or smooth muscle cells [95,96,97]. The N-terminal domain of THBS1 has been demonstrated to be involved in promoting angiogenesis induced by the apoptosis protective ability of endothelial cells [97,98]. Furthermore, platelets secrete THBS1, an inflammatory mediator that is necessary to induce angiogenesis and neurorepair during cerebral ischemia [68,93,94,99]. Lastly, *THBS1* gene silencing induced by DNA methylation in post-stroke recovery inhibits neurorestoration, thus exacerbating stroke injury [7,68,72,100]. 

The role of MBD2 in promoting angiogenesis and vascular endothelial cell survival by acting on vascular endothelial cells has also been addressed [101]. Changes in the DNA methylome are read by a family of proteins that share MBDs (i.e., MBD1, MBD2, MBD3, MBD4, and MeCP2) [101,102,103,104], and are common in humans and experimental animal models of vascular disease [101,105,106]. One of the important roles of MBD proteins is to maintain and directly interact with the DNA methylome, which is involved in the DNA methylation-mediated repression of gene expression and/or heterochromatin formation [101]. The effect of DNA methylation on endothelial function in ischemic stroke was also analyzed on the basis of the regulation of MBD2, an interpreter of the information-encoding DNA methylome [101]. MBD2 has been demonstrated to be involved in apoptosis after changes in oxidative stress and cell proliferation in HUVECs subjected to *MBD2* knockdown. Consequently, *MBD2* knockdown significantly enhances angiogenesis and protects endothelial cells from hydrogen-peroxide-induced apoptosis. Experimentally, proliferation is significantly improved in *MBD2*-knockdown cells compared with that in cells transfected with control siRNA, and apoptosis induced by hydrogen peroxide is inhibited. These results suggest that MBD2 negatively regulates angiogenesis and renders vascular endothelial cells more responsive to oxidative damage-induced apoptosis. Endothelial cell survival and angiogenesis signaling are activated by MBD2 inhibition. After *MBD2* knockdown, activation of the extracellular signal-regulated protein kinase (Erk)1/2 signaling factor, which is involved in endothelial cell survival, is significantly increased, and the expression of B-cell lymphoma 2 (BCL-2), a downstream pathway of Erk1/2, is also increased to suppress apoptosis stimulation. Consistently, *Mbd2*-knockout mice subjected to femoral artery resection show significantly improved perfusion accompanied by increased capillary and arteriole formation [101]. Overall, these findings indicate that inhibition of MBD2 activity has the potential to improve angiogenesis, which is a major mechanism in neural recovery after ischemic stroke.

### 5.3. Axonal Growth

Axonal growth after nerve injury is an important recovery process for restoring the plasticity and dysfunction caused by injury. The functional significance of axonal regrowth is dependent on several neurobiological adjustments, including myelination, formation of synapses, and abnormal interconnection pruning [107,108]. Recent findings have elucidated the epigenetic mechanisms underlying enhanced axonal regeneration [109,110]. Specifically, the importance of DNA methylation in axonal regeneration has been significantly advanced in recent years [110,111,112]. However, the role of DNA methylation in axonal regeneration in ischemic stroke models remains poorly explored. 

In a study on alteration of axonal regeneration by inhibition of DNMT activity, it was found that axonal regeneration after sciatic nerve crush injury in mice was found to be inhibited by treatment with the nonnucleoside DNMT inhibitor RG108 (*N*-phthalyl-l-tryptophan), suggesting that DNA methylation is necessary for axonal regeneration [111].

Furthermore, Nogo-A (reticulon 4), a representative growth inhibitory factor, is widely expressed in the mammalian CNS. Neuronal regeneration is induced by inhibition of Nogo-A or blockade of the Nogo-A pathway [113,114]. After binding to the Nogo receptor, Nogo-A stimulates Ras homolog family member A (RhoA)/Rho-associated coiled-coil-containing protein kinase 2 (ROCK2), which is involved in downstream signaling pathways, to inhibit axonal growth and interfere with nerve repair and regeneration [113,115,116]. Therefore, blocking the complete Nogo-A/RhoA/ROCK pathway is necessary to remove large obstacles that impede the neural regeneration process after ischemic stroke. In a recent study on the efficacy and mechanism of the traditional Chinese medicine naoluoxintong (NLXT) in ischemic stroke, expression of the *Nogo-A/RhoA/ROCK* genes, which was significantly upregulated in the ischemic stroke model, was found to be significantly reduced by NLXT treatment. The downregulation of Nogo-A, NgR1, and RhoA was reversed by treatment with the DNMT blocker SGI-1027. This suggests that reduced DNA methylation induced by DNMT inhibition affects the expression of Nogo-A, NgR1, and RhoA [113].

Recent experimental evidence indicated that axonal resection of mouse dorsal root ganglion (DRG) neurons resulted in elevated levels of methylcytosine dioxygenase 3 (TET3) [112,117]. Moreover, 5mC is converted to 5hmC by TET3, followed by repeated oxidation to 5fC and 5caC, which catalyzes DNA demethylation. Whole-genome mapping of 5hmC identified distinct changes in activating transcription factor 3 (*Atf3)*, brain-derived neurotrophic factor (*Bdnf)*, and SMAD family member 1 (*Smad1)* genes, which are involved in regeneration after brain injury [117,118]. *Tet3* knockdown significantly reduced the axonal growth capacity of DRG neurons; furthermore, after injury, methylation of CpG dinucleotides was significantly reduced in both the gene body and the enhancer region of *Atf3* [112,118]. Furthermore, the role of thymine DNA glycosylase (Tdg), which is involved in regeneration-associated gene (RAG) expression and axon regeneration, was also elucidated. Tdg acts on *Tet3* subprocesses by removing 5fC and 5caC to initiate base cleavage repair and produces unmodified cytosine. Inhibition of Tdg reduces RAG expression, together with attenuated axonal regeneration. Conversely, the axonal regeneration induced after phosphatase and tensin homolog *(Pten)* deletion in mouse RGCs is attenuated by the inhibition of TET1, but not TET3 [112,118]. Therefore, axonal regeneration induced after neuronal injury may be promoted by regulating the enzymes involved in removing the methyl groups after DNA methylation.

In addition to the previously described increase in the expression of TET-dependent RAGs or the promotion of axonal regeneration by DNA methylation, glial scar formation increases the production of endogenous growth inhibitory agents that appear during the growth of newly regenerated axons [72,108,119]. Factors that interfere with axonal growth accumulate in the vicinity of cerebral infarction after such events and must be controlled for axonal regeneration. Currently, the role of DNA methylation in the regulation of glial scar formation and the endogenous growth inhibitory factors involved in regulating axial firing after stroke remain unexplored. Interestingly, the expression of the small proline-rich protein 1 (SPRR1), which promotes axonal growth, is reportedly increased after stroke [72,108,120,121], and hypomethylation induced by treatment with 5-azacytidine, a DNMT inhibitor, in keratinocytes, leads to rapid upregulation of SPRR1 [122].

We recently demonstrated the effect of modulating DNA methylation on recovery from the chronic phase in a motor cortex injury stroke model [123]. Although the role of motor function recovery by task-oriented training after stroke is well known, this effect is limited aa a function of the period following ischemic stroke onset. Therefore, DNA methylation control and task-oriented training were performed concomitantly to overcome the limitation of recovery in the chronic phase. When 5-aza-dC, a DNMT inhibitor, was administered together with task-oriented training, the level of impaired motor function after stroke was improved significantly [123]. Analysis of the underlying mechanism revealed that axonal growth from the contralateral cortex to the corticospinal cord was increased, along with the maturation of BDNF. This implies that regulation of DNA methylation in the chronic phase with limited recovery after stroke can help resume the recovery process in the acute phase; thus, regulating DNA methylation after stroke has potential for development as a new treatment technique.

### 5.4. Synaptic Plasticity

Synaptic plasticity is the primary factor in the formation and preservation of neural nets and is an important mechanism in the stroke-related natural recovery process [77,78]. Significant functional cortical remapping occurs in both animal models and human patients after stroke [72,124,125]. Several weeks after a stroke, these changes are attributed to the increased expression of synaptic proteins in both the peri-infarct regions of the ipsilateral cortex and the contralateral cortex in the brain [72,126].

The synaptic plasticity occurring during stroke recovery depends on epigenetic regulation [72,127]. After stroke onset, the acute inflammatory process alters the level of DNA methylation, which may play a significant role in ischemic injury and recovery [77]. Notably, induction of long-term synaptic plasticity is linked to both DNA methylation and the MBD [127,128]. Experimental studies have revealed an association between DNA methylation and the Hebbian theory of synaptic plasticity, which involves mechanisms associated with memory storage [127]. The significance of DNMTs in modulating synaptic plasticity mechanisms was also investigated. DNMT1 and DNMT3a levels are elevated in differentiated hippocampus neurons [72,129,130,131], suggesting that DNA methylation plays a major role in synaptic transmission and control during the learning and memory processes in hippocampal neurons [129]. According to recently published findings, DNMT inhibition in normal hippocampal neurons affects neural excitability and network activity after decreasing the frequency of miniature excitatory postsynaptic currents, together with decreased methylation, which inhibits genomic DNA activity [129,132]. Furthermore, long-term memory and synaptic plasticity were found to be blocked by DNA methylation inhibition in hippocampal slice experiments [133]. Furthermore, hippocampal synaptic plasticity was found to be enhanced in mice deficient in Gadd45b, an active DNA demethylation regulator, enhancing hippocampal-dependent memory and memory consolidation [134]. 

In a study on DNA methylation associated with memory formation in an animal model of fear conditioning, *Dnmt* gene expression was upregulated in the hippocampus of adult rats that received contextual fear regulation, and memory formation was blocked after administering a DNMT inhibitor [132,133,135]. Fear conditioning results in methylation of type I protein phosphatase (PP1), which acts as a memory suppressor, leading to transcriptional silencing, whereas demethylation of the synaptic plasticity gene, *reelin*, activates transcription [135]. Both methyltransferase and demethylase activities are observed during memory consolidation [135]. DNMT inhibition avoids an increase in PP1 methylation, which results in aberrant gene transcription during the consolidation period of memory [135]. Overall, these findings suggest that the adult nervous system dynamically regulates DNA methylation, and that memory formation is strongly influenced by regulation of DNA methylation [135]. Therefore, sustaining DNMT activity in mature neurons is essential for synaptic function [129].

In recent research, sustained depolarization of primary cortical neurons in vitro was shown to reduce methylation in the promoter region of *Bdnf*, which encodes a neurotrophin crucial for synaptic plasticity [77,132,136]. Increased BDNF expression and mature BDNF protein were also observed in task-trained stroke mice treated with the DNMT inhibitor 5-aza-dC. This therapeutic effect enhances the neuroplasticity of the corticospinal tract, which controls the motor function of forelimbs [123]. DNA methylation is a major epigenetic regulatory mechanism for *Bdnf* gene transcription [77,137,138]. The *BDNF* gene contains one 3′ coding exon, which is responsible for encoding the amino-acid sequence of the BDNF pre-protein, and nine 5′ noncoding exons (I–IXa), each associated with a separate promoter region [137,139]. In the hippocampus of depressed mice after stroke, DNA methylation was significantly increased in the specific CpG region of the *Bdnf* promoter IV, and MeCP2 was bound to the methylated *BDNF* promoter, resulting in a repressor complex, which prevented the cAMP response element (CRE) from binding to the CRE-binding protein (CREB) and suppressed *Bdnf* gene transcription [137].

The mechanism underlying the pharmacological effect of fluoxetine in treating poststroke depression involves upregulation of BDNF by separating the MeCP2–CREB–BDNF promoter IV complex via protein kinase A (PKA)-induced phosphorylation of MeCP2 at Ser421 [137]. This suggests that fluoxetine regulates the BDNF expression involved in synaptic plasticity through epigenetic mechanisms.

As noted, the transient DNA methylation in hippocampal neurons required for memory formation can be controlled by either DNMT1 or DNMT3a [72,140]. In this regard, demethylase activity is important as it regulates the expression of specific plasticity-related genes in the mature brain [135,141]. Neuronal activation by electroconvulsive stimulation (ECS) induces DNA demethylation, which increases cell proliferation and synaptic plasticity [141,142]. After stroke, neurons in the peri-infarct cortex become hyperexcited [72]. This enhanced excitatory neuronal activity may be associated with dynamic modulation of the DNA methylation state, as in the activation of neurons by ECS [21,72,141]. Increased neural activity was shown to enhance the selective demethylation of BDNF by activating Gadd45b and TET1 [21,72]. Therefore, the mechanism of demethylation to regulate neural activity may be crucial for stroke recovery.

## 6. Results and Discussion

This review aimed to investigate the role of DNA methylation in the pathological processes after stroke and to confirm the possibility of applying its modulation as a treatment for stroke. Studies on DNA methylation changes in excitotoxicity, oxidative stress, mitochondrial dysfunction, BBB disruption, apoptosis, and inflammatory mechanisms after cerebral infarction were reviewed for application to neuroprotective mechanisms after stroke [2,3,4,31]. In addition, we also assessed studies examining the effect of DNA methylation changes on neurogenesis, angiogenesis, axonal plasticity, and synaptic plasticity related to nerve recovery after stroke [5,6]. 

Table 1 summarizes the content of 26 selected studies on DNA methylation levels, target genes, DNA methylation regulators, and DNA methylation regulatory mechanisms that cause neuronal damage and promote neuronal recovery after stroke.

The *EAAT2* gene promoter in astrocytes is hypermethylated because of increased DNMT1 and DNMT3a activity in the excitotoxicity appearing after stroke [35], along with decreased methylation of the *NKCC1* promoter [9,27]. MBD2 binds to the *EAAT2* gene promoter with increased methylation to regulate gene transcription [35].

In oxidative damage, global DNA methylation is increased owing to increased DNMT1 activity [8], and demethylation occurs via TET3 [47]. Increased expression of neuroprotective genes after ischemic stroke was induced by an increase in 5hmC levels by TET3 in the gene promoter region [47].

DNA methylation changes related to mitochondrial dysfunction show increased *Timp-2* methylation and decreased *Mmp-9* methylation through regulated expression and activity of DNMT1 and DNMT3a [48]. In demethylation of mitochondrial DNA, the level of mitochondrial TET2-mediated 5hmC increased after ischemic brain injury, resulting in increased expression of six mitochondrial genes (ND2, ND3, ND4L, ND5, ND6, and COX3) [51].

BBB disruption is associated with *TIMP-2* hypomethylation and MMP-9 hypermethylation via DNMT3b regulation [48,54].

In neuronal cell death including apoptosis after stroke, DNMTs, DNMT1, and DNMT3a expression regulates increased or decreased global DNA methylation [8,9,12,26,58,143]. Fetal hypoxia induced a decrease in GR expression by increased DNA methylation in the hippocampus of neonatal mice, as well as increased susceptibility of the brain to hypoxia–ischemic injury [60]. The expression of neuroprotection-related genes Arid5a, Nptx2, and Stc2 was increased via hypomethylation of DNA in ischemic pretreatment mice [61]. In addition, after MCAO, mRNA expression of estrogen receptor alpha (ERα), known to have neuroprotective effects, is upregulated in the cortex of female mice. This mechanism is associated with reduced methylation of the promoter region of the ERα gene and inhibition of MeCP2 binding to methylated DNA [7,63,64]. In the demethylation mechanism, upregulation of 5hmC by TET1, TET2, and TET3 has a protective effect against neuronal cell death caused by oxidative stress and apoptosis in ischemic stroke and neurodegenerative diseases [47].

In the neuroinflammatory response, hypomethylation of *LINE-1* in human blood increases the incidence of stroke [9]. Furthermore, increased expression of *TRAF3* [30], *PPM1A* [71], and *MTHFR* [8] because of DNA hypomethylation results in increased activity of neuroinflammation factors. In the PST model, reduced demethylation in the *Lef1* gene promoter by downregulation of TET2 decreased the expression of Lef1 mRNA and increased the expression of inflammatory factors [76].

Among the mechanisms of neuronal recovery after stroke, regulation of neurogenesis-inducing DNA methylation induces hypermethylation of *galanin* and *Ng2* gene promoters via DNMT upregulation, and global DNA methylation is also increased [81]. However, demethylation also in the *galanin* and *Ng2* promoter regions increases with increasing TET1 [79]. Gadd45b induced demethylation at the DNA promoter regions of *Bdnf* and *Fgf-1*, resulting in increased expression of Bdnf and fgf-1 mRNA [72].

In angiogenesis-related DNA methylation studies, both increased and decreased of DNA methylation can regulate the antiangiogenic THBS1 and promote angiogenesis [8,71].

Changes in DNA methylation that increase axonal regrowth and plasticity, which are important for post-stroke recovery, involve either reduced global hypermethylation upon treatment with DNMT inhibitors [123] or increased methylation of the promoters of *NogoA* and *Ng2* genes [113], which encode factors that inhibit axonal growth. In addition, decreased methylation in the gene promoter of the regeneration-related factor SPRR1 [72] can increase axonal regrowth, and upregulation of TET3 can induce demethylation, thereby increasing the transcription of regeneration-related genes (RAG) [47].

Increased synaptic plasticity is involved in global hypermethylation-induced memory formation via the upregulation of DNMTs, DNMT1, and DNMT3a in the hippocampus [72]. In contrast, reduction in methylation induced by DNMT inhibitor treatment after stroke induces enhanced corticospinal repair [123] and increased BNDF expression [77]. Increased demethylation by TET1 [72] and Gadd45b [72], and decreased MeCP2 binding [137] in the *Bdnf IV* promoter increases BDNF expression and enhances synaptic plasticity and neuronal activity.

The involvement of DNA methylation in stroke pathophysiology, including stroke risk factors, pathogenesis, and post-onset damage and recovery, reveals dynamic changes depending on the injury-related environment. In general, the degree of DNA methylation and demethylation in the promoter regions of global genes or selected genes is regulated by DNMT and TET, respectively. As DNA methylation differs with each stage, process, and function after stroke, additional research and development are needed to better understand these factors and enable more selective control.

As discussed in Section 4 and Section 5, various studies on neuroprotection and neurorepair after stroke by regulation of DNA methylation are in progress, but more advanced studies are needed. By including a discussion of the study results of stroke-related diseases and non-neuronal cells, the goal was to contribute to expanding the applicability of research on stroke treatment. The presented factors regulating DNA methylation, which are involved in neuroprotection and neurorepair mechanisms after stroke, are expected to contribute to the development of new strategies for stroke treatment and the development of potential therapeutic agents.

## 7. Conclusions

Patients with stroke worldwide suffer from long-term disability. Despite the active use of tPA therapeutics, the treatment period is narrow and there are side-effects; therefore, more effective therapeutics need to be developed [1,2,3,4].

For the past 20 years, studies on DNA methylation regulation among epigenetic mechanisms affecting stroke induction, treatment, and recovery have been actively conducted. DNA methylation increases or decreases according to changes in the expression and activity of DNMT. Removal of DNA methylation results in conversion to 5hmC, 5-fc, and 5-cac by TET activity, and gene transcription is regulated by the binding of MBD family proteins to the methylated CpG sites [9,22,24].

In the results of studies analyzed in this review, the roles of hypermethylation or hypomethylation of all or specific genes induced by DNMT modification after stroke, the role of 5hmC demethylation induced by TET, and the role of association or dissociation of MBP on gene expression were highlighted. The importance of these mechanisms in stroke was elucidated through experiments that inhibit or eliminate the expression of DNMT, TET, and MBP. Alterations of DNA hypermethylation, hypomethylation, demethylation, and MBP binding are induced in post-stroke-induced excitotoxicity, oxidative stress, mitochondrial dysfunction, BBB dysfunction, apoptosis, and neuroinflammatory mechanisms. Therefore, neuroprotection in stroke is achieved using reversible regulation of changed DNA methylation after a stroke.

Hypermethylation and hypomethylation by changes in DNMT, DNMT1, DNMT3a, and DNMT3b modulate the global genes or expression of *EAAT2*, *NKCC1*, *TIMP-2*, and *MMP-9* genes. Demethylation by TET1, TET2 and TET3 modulates the expression of genes related to neuroprotection, mitochondrial subunits, and inflammatory regulators in stroke and alters *EAAT2* and *ERα* expression by regulation of MBD2 and MeCP2.

In the mechanisms of recovery after stroke, including neurogenesis, angiogenesis, axonal growth, and synaptic plasticity, via regulation of DNMTs, DNMT1, and DNMT3a, hypermethylation or hypomethylation is induced not only in the global gene promotor but also in the promotor regions of *galanin*, *Ng2*, *THBS1*, *SPRR1*, and *BDNF* genes. Demethylation by TET1, TET3, and Gadd45b controls the expression of galanin, *Ng2*, *bdnf*, *fgf-1*, RAG, and *Bdnf IV* in the neurorepair mechanism in stroke. In addition, by reducing the binding of MeCP2 to the methylated DNA of the *Bdnf IV* promoter, BDNF expression is increased in the synaptic plasticity in stroke recovery.

This review can help researchers understand the recent experimental and knowledge-based evidence for the regulation of DNA methylation in neuroprotective and neuronal repair mechanisms after stroke.

Furthermore, the factors presented in this review may be applied to the development of novel treatment strategies and potential therapeutics for stroke treatment.

## Figures and Tables

**Figure 1 ijms-23-10373-f001:**
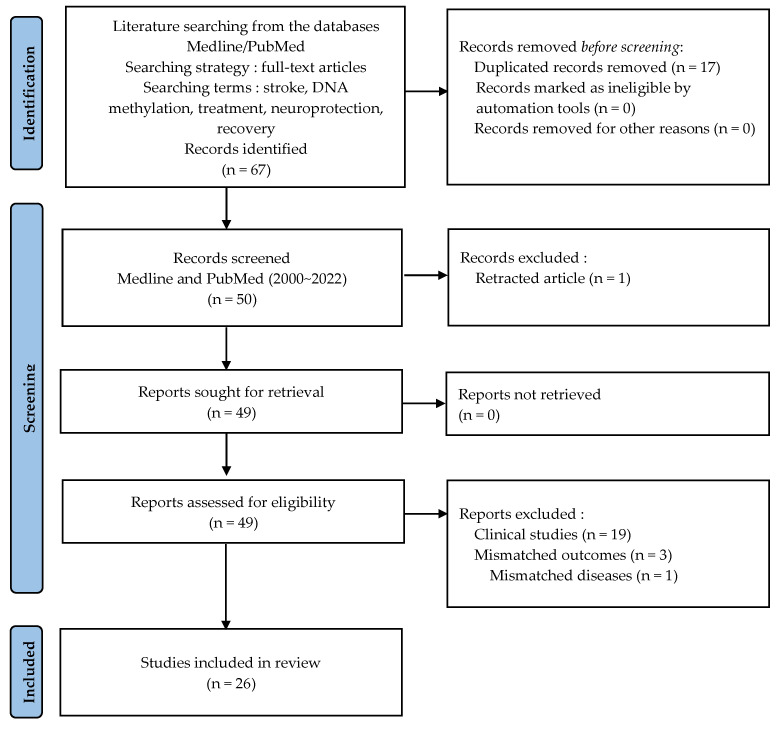
The flowchart of database searching, screening, selection, and inclusion of eligible articles from the literature.

**Table 1 ijms-23-10373-t001:** Alterations of DNA methylation involved in neuroprotection and neurorepair mechanisms after stroke.

Therapeutic Strategy	Mechanisms of Pathology or Repair Processes	Alteration of DNA Methylation	Target Gene	Regulating Factors	Upregulation or Downregulation	References
Neuroprotection	Excitotoxicity	Hypermethylation	*EAAT2*	DNMT1, DNMT3A	Up	[35]
		Hypomethylation	*NKCC1*	DNMT	Down	[9,27]
		Binding to methylated CpG	*EAAT2*	MBD2	Down	[35]
	Oxidative stress	Hypermethylation	Global	DNMT1	Up	[8]
		Demethylation	Global and neuroprotective genes	TET3	Up	[47]
	Mitochondrial dysfunction	Hypermethylation	*TIMP-2*	DNMT1, DNMT3a	Up	[48]
		Hypomethylation	*MMP-9*	DNMT1, DNMT3a	Down	[48]
		Demethylation	*ND2, ND3, ND4L, ND5, ND6, and COX3*	TET2	Up	[51]
	Blood–brain barrier disruption	Hypermethylation	*Timp-2*	DNMT3b	Up	[48,54]
		Hypomethylation	*MMP-9*	DNMT3b	Down	[48,54]
	Apoptosis	Hypermethylation	Global	DNMT1, DNMT3a DNMTs	Up	[8,9,12,26,58,143]
			*GR*	DNMTs	Up	[60]
		Hypomethylation	*Arid5a, Nptx2, Stc2*	ND	ND	[61]
			*ERα*	ND	ND	[7,63,64]
		Demethylation	Global	TET1, TET2, TET3	Up	[47]
		Binding to methylated CpG	*ERα*	MeCP2	Up	[7,63,64]
	Inflammation	Hypomethylation	*LINE-1*	ND	ND	[9]
			*TRAF3*	ND	ND	[30]
			*PPM1A*	ND	ND	[71]
			*MTHFR*	ND	ND	[7]
		Demethylation	*LEF1*	TET2	Down	[76]
Neurorepair	Neurogenesis	Hypermethylation	Global, *Galanin, Ng2*	DNMT	Up	[81]
		Demethylation	Global, *Galanin, Ng2*	TET1	Up	
			*Bdnf, Fgf-1*	Gadd45b	Up	[72]
	Angiogenesis	Hypermethylation	*THBS1*	DNMT	Up	[7,71]
		Hypomethylation	*THBS1*	DNMT	Down	[7,71]
	Axonal growth	Hypermethylation	*NogoA, NgR*	DNMT	Up	[113]
		Hypomethylation	Global	DNMT	Down	[123]
			*SPRR1*	DNMT	Down	[72]
		Demethylation	RAG	TET3	Up	[47]
	Synaptic plasticity	Hypermethylation	Global	DNMTs, DNMT1, DNMT3a	Up	[72]
		Hypomethylation	Global	DNMTs	Down	[123]
			*BDNF*	DNMTs	Down	[77]
		Demethylation	*Bdnf* IV	TET1	Up	[72]
				Gadd45b	Up	[72]
		Binding to methylated CpG	*Bdnf* IV	MeCP2	Down	[137]

ND: not determined.

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
