# Peer review of "The Role of DNA Methylation in Stroke Recovery"

_ijms, 2022, doi:10.3390/ijms231810373_

Round 1

Reviewer 1 Report

I read through the manuscript, it's an easy read. The review is well written and comprehensively discussed. Albeit a bit too comprehensive for a mini-review. Good luck to the authors for the future.

Reviewer 2 Report

the paper is interesting, although the techniques used in the systematic review are not appreciated.

I suggest pointing out the techniques used in methodology.

I suggest that the conclusions be from the review, and not suggestions.

I suggest a reduction of the article, since it has to occupy half the pages. best regards.

Round 2

Reviewer 2 Report

the authors have made the suggested changes.

I suggest the following reference.

10.3390/jcm8101712